# Incidence of Isolated Biliary Atresia during the COVID Lockdown in Europe: Results from a Collaborative Project by RARE-Liver

**DOI:** 10.3390/jcm12030775

**Published:** 2023-01-18

**Authors:** Mark Nomden, Naved K. Alizai, Pietro Betalli, Janneke L. M. Bruggink, Mara Cananzi, Vibeke Brix Christensen, Lorenzo D’Antiga, Mark Davenport, Björn Fischler, Luise Hindemith, Maria Hukkinen, Lars S. Johansen, Ruben H. de Kleine, Omid Madadi-Sanjani, Evelyn G. P. Ong, Mikko P. Pakarinen, Claus Petersen, Mathias Ruiz, Matthias Schunn, Ekkehard Sturm, Henkjan J. Verkade, Barbara E. Wildhaber, Jan B. F. Hulscher

**Affiliations:** 1Division of Paediatric Surgery, Department of Surgery, University of Groningen, University Medical Centre Groningen, 9713 GZ Groningen, The Netherlands; 2European Reference Network for Hepatological Diseases (ERN RARE-LIVER), D-20246 Hamburg, Germany; 3Department of Paediatric Surgery & Liver Unit, Leeds General Infirmary, Leeds LS1 3EX, UK; 4Department of Paediatric Surgery, ASST Papa Giovanni XXIII, 24127 Bergamo, Italy; 5Pediatric Gastroenterology, Digestive Endoscopy, Hepatology and Care of the Child with Liver Transplantation, Department of Women’s and Children’s Health, University Hospital of Padova, 35128 Padova, Italy; 6Department of Pediatrics and Adolescent Medicine, Copenhagen University Hospital, Rigshospitalet, DK2100 Copenhagen, Denmark; 7Department of Paediatric Hepatology, Gastroenterology and Transplantation ASST Papa Giovanni XXIII, 24127 Bergamo, Italy; 8Department of Paediatric Surgery, Kings College Hospital, London SE5 9RS, UK; 9Paediatric Digestive Diseases, Astrid Lindgren Children’s Hospital, CLINTEC, Karolinska Institutet, Karolinska University Hospital, 171 64 Stockholm, Sweden; 10Department of Paediatric Surgery, Hannover Medical School, 30625 Hannover, Germany; 11Paediatric Liver and Gut Research Group, Children’s Hospital, Helsinki University Hospital, University of Helsinki, 00029 Helsinki, Finland; 12Section of Paediatric Surgery, Children’s Hospital, Helsinki University Hospital, University of Helsinki, 00029 Helsinki, Finland; 13Department of Pediatric Surgery, Copenhagen University Hospital, Rigshospitalet, DK2100 Copenhagen, Denmark; 14Division of Hepatico-Pancreatico-Biliary Surgery and Liver Transplantation, Department of Surgery, University of Groningen, University Medical Centre Groningen, 9713 GZ Groningen, The Netherlands; 15The Liver Unit, Birmingham Children’s Hospital, Birmingham B4 6NH, UK; 16Department of Paediatric Gastroenterology, Hepatology and Nutrition, Reference Center for Rare Disease–Biliary Atresia and Genetic Cholestasis, Children’s Hospital of Lyon, 69005 Lyon, France; 17Department of Paediatric Surgery and Paediatric Urology, University Children’s Hospital, 72076 Tübingen, Germany; 18Paediatric Gastroenterology/Hepatology, University Hospital Tübingen, 72076 Tübingen, Germany; 19Division of Paediatric Gastroenterology and Hepatology, Department of Paediatrics, Beatrix Children’s Hospital, University of Groningen, University Medical Centre Groningen, 9713 GZ Groningen, The Netherlands; 20Swiss Pediatric Liver Centre, Division of Pediatric Surgery, University Hospitals of Geneva, Department of Pediatrics, Gynecology, and Obstetrics, University of Geneva, 1205 Geneva, Switzerland

**Keywords:** biliary atresia, COVID lockdown, stringency index, European reference network RARE-LIVER, epidemiology

## Abstract

Background: Biliary atresia (BA) is a rare cholangiopathy where one of the proposed aetiological mechanisms is an infectious viral trigger. Coronavirus disease-19 (COVID) lockdown restrictions were implemented to reduce the transmission of infections. Strictness of lockdown varied across European countries. This study aimed to investigate if there was an association between strictness of lockdown and change in isolated BA (IBA) incidence in Europe. Methods: We approached European centres involved in the European Reference Network RARE-LIVER. We included IBA patients born between 2015 and June 2020. We calculated the number of IBA patients born per centre per month. The Stringency Index (SI) was used as lockdown strictness indicator. The association between percentage change of mean number of IBA patients born per month and the SI was assessed. Results: We included 412 IBA patients from thirteen different centres. The median number of patients per month did not change: 6 (1–15) pre-lockdown and 7 (6–9) during lockdown (*p* = 0.34). There was an inverse association between SI and percentage change in IBA (B = -0.73, *p* = 0.03). Median age at Kasai portoenterostomy (days) did not differ between time periods (51 (9–179) vs. 53 (19–126), *p* = 0.73). Conclusion: In this European study, a stricter COVID-lockdown was seemingly accompanied by a simultaneous larger decrease in the number of IBA patients born per month in the lockdown. Results should be interpreted with caution due to the assumptions and limitations of the analysis.

## 1. Introduction

Biliary atresia (BA) is a rare disease of unknown origin. One hypothesis suggests that the pathophysiology involves a viral infection, followed by an over-activation of the immune response aimed at cholangiocytes [1,2]. This type of BA is termed isolated BA (IBA) to distinguish it from the syndromic form of BA, which is associated with developmental errors [2]. Prime viral candidates as the initiators of immune overactivation in IBA are rotavirus and cytomegalovirus (CMV) [3,4,5]. However, an unambiguous causal relationship between a virus and IBA has not been demonstrated yet, although CMV-IgM positivity is considered to define a separate subtype of BA [6,7,8]. Although a popular hypothesis, due to the lack of convincing evidence, the possibility of developmental and/or genetic factors playing a role in the pathophysiology of IBA cannot be excluded [9].

In February 2020, Europe was affected by the global pandemic caused by the Severe Acute Respiratory Syndrome Coronavirus 2 (SARS-CoV-2 or COVID-19) [10]. As a consequence, many European countries took protective measures to reduce the exposure of the general population to people infected with COVID-19. Initial measures consisted of measures increasing personal hygiene, such as frequent hand washing with alcohol-based hand sanitiser or soap, and social measures such as social distancing oneself in order to minimise the risks of contracting infectious droplets when a person sneezes or speaks [11,12,13]. In March 2020, more drastic measures to reduce the transmission of COVID-19 were taken. Most European countries went into various degrees of lockdown, meaning closure of public primary and secondary schools, day-care centres as well as all public places such as restaurants and sport clubs. Generally, this first strict lockdown lasted from March 2020 up to and including May 2020 in most European countries. Although aimed to reduce COVID-19-related transmission, the transmission of non-COVID-19 viruses and bacteria was also affected by these strict measures. For example, in the Netherlands, a sharp decrease in the number of paediatric infectious diseases such as the common flu, otitis media and gastroenteritis was observed [14].

We hypothesised that these lockdowns might have had an effect on the incidence of IBA, thereby offering further support for an (infectious) environmental origin of this still enigmatic disease. Strictness of lockdown was not similar across Europe, with various measures applied in different countries. We therefore hypothesised that the strictness of lockdown would be associated with a decrease in IBA incidence, with a stricter lockdown leading to a more pronounced decrease in IBA incidence. Therefore, we performed a collaborative European-wide study that aimed to investigate the incidence of IBA during the lockdown period in different European countries and to assess the association between strictness of lockdown and IBA incidence.

## 2. Materials and Methods

We invited members of the European Reference Network RARE-Liver and the Biliary Atresia and Related Disorders (BARD) community to provide data on IBA patients treated in their centres prior to and during the COVID pandemic. The ERN RARE-Liver is a Europe-wide network for centres of expertise in the clinical management of rare liver diseases, for example BA and choledochal malformation. BARD is a collaborative initiative to perform international research on BA and related diseases. This study was performed in accordance with the guidelines of the Medical Ethical Committee of the University Medical Centre Groningen (2017/056).

We collected patient data of BA patients born between 2015 and 2021 via a REDCap online database [15]. Clinical variables were date of birth, place of birth and age at primary treatment (Kasai portoenterostomy (KPE) or primary liver transplantation). Patients were excluded for this study if (1) they did not have the isolated variant of BA, (2) if their primary treatment did not take place in the participating centre (KPE or primary liver transplantation) or (3) if they were born outside of the country in which the participating centre is located. Patients were considered as non-isolated according to when they had biliary atresia splenic malformation (BASM) syndrome or a congenital abnormality that is associated with syndromal BA [9].

Strictness of lockdown was graded according to the ‘stringency index’ (SI). The SI is an open access index calculated based on multiple variables such as whether schools were closed during the lockdown or how strict the stay-at-home advice was. Each variable is scored separately from 0 to 100 from which an overall SI is calculated per country [16]. We selected items of the SI that we deemed relevant for the transmission of bacteria and viruses in children. Six items were included: school closing, stay at home advice, restrictions on public gatherings, restriction on public transport, cancellation of public events and closure of workplace.

The score of each included variable was calculated as described by Hale et al. [16]. Briefly, each included variable received a grade based on an ordinal scale every day during the lockdown period. The lockdown period was defined as the period from March 2020 up to and including May 2020. The ordinal grade was transformed to a score between 0 (no restriction) and 100 (maximum restriction) for every individual day. Subsequently, the mean score of each individual variable during the lockdown period was calculated. SI was calculated by averaging the mean score of each individual included variable during the lockdown period.

SI is an indirect measure of viral transmission and a higher SI does not necessarily indicate a more efficient response. SI is correlated to COVID-related deaths and, although still a surrogate marker with accompanying limitations, was therefore deemed an acceptable measure of strictness of lockdown and viral transmissions [17]. Furthermore, the daily number of COVID infections was limited by insufficient testing capacity, fluctuations and variable testing policy in different countries and was therefore unsuitable as surrogate marker.

Two time periods were defined: the period prior to the lockdown (i.e., 1 January 2015–29 February 2020) and the period of the strict lockdown (1 March 2020–31 May 2020). Date of birth of patients who presented with IBA was used to determine the number of patients born per month who acquired IBA per centre. Based on the date of birth, patients were either classified as born in the pre-lockdown period or the lockdown period. 

Treatment for BA in a country can be centralised or non-centralised. When treatment of BA is centralised, BA patients are treated in one or a few designated centres. When treatment is non-centralised, multiple centres can treat BA patients with variable caseloads. For countries with centralised BA care, national data on all BA patients can be reliably obtained. When not all centres from a country where BA treatment is centralised participate or when centres from countries with non-centralised BA treatment participate, only a part of the national BA data can be obtained.

Incidence of IBA can only be calculated in countries with complete, national BA data because a reliable population at risk can be determined (i.e., live births). When only part of the national BA data is present, incidence cannot be calculated. Therefore, participating centres were divided into two groups based on whether national data or a part of the national data were present: national BA data and part of national BA data. When complete national BA data were available, country name is reported. When part of the national BA data was available, centre name is reported.

The median number of IBA patients born per month for all centres was combined and each centre (part of national data) or country (national data) was recorded separately. Mean number of IBA patients born per month was calculated. The percentage change in mean number of IBA patients born per month during the pre-lockdown period and lockdown period was calculated. This was completed for total IBA patients, per centre and per country.

For countries with national BA data, incidence of IBA was calculated. Incidence was calculated by dividing the number of IBA patients by the number of live births per country, multiplied by 10,000. Data on live births were retrieved from the national bureau of statistics from each respective country. Centres from the United Kingdom, the Netherlands, Finland, Switzerland and Denmark had complete national BA data and were therefore included in this analysis. IBA incidence in the two time periods was compared.

Clustering of IBA in time is only found in some studies [18,19,20,21,22,23,24,25,26,27,28]. However, to avoid any possible clustering in time to affect our analysis, we also compared the incidence of IBA during the lockdown period in 2020 and the IBA incidence in the three-month period of March–June in the years 2015 up to and including 2019.

To detect a potential delay in referral during the strict lockdown, we compared median age at time of KPE (excluding patients that were treated by a primary liver transplantation) in the period prior to the lockdown and in the lockdown of patients with IBA.

### Statistical Analysis

Difference in the median number of IBA patients born per centre per month and age at KPE (when KPE was primary treatment) during the pre-lockdown and lockdown period was compared using the Mann–Whitney *U* test.

Calculated SI was treated as a continuous variable. The association between the percentage change in the mean number of IBA patients born per centre per month between each time period and SI was assessed using Spearman’s correlation coefficient. Percentage change was calculated as follows:Percentage change = (mean IBA patients born per month in lockdown–mean IBA patients born per month in pre-lockdown)/mean IBA patients born per month in pre-lockdown.(1)

To evaluate differences in incidence (i.e., national data), the incidence of IBA in the period prior to the lockdown was used as a reference category. IBA incidence during the lockdown was compared by using logistic binominal regression analysis and an odds ratio (OR) was displayed. We evaluated the association between SI and IBA incidence during the lockdown by using multivariate logistic regression. BA was used as a binominal dependent outcome, with SI as a continuous independent variable. Country was included as a categorial independent variable.

Statistical analysis was performed using IBM SPSS statistical software for Windows, version 23.0. (IBM corp, Armonk, NY, USA) and Stata Statistical Software: Release 14 (StataCorp LP, College Station, TX, USA). Graphs were created using GraphPad Prism version 8.4.2 for Windows (GraphPad Software, San Diego, CA, USA).

## 3. Results

Thirteen centres from eight countries participated in this study, thereby including 412 patients. A total of 95 patients were classified as having a non-isolated form of BA and were excluded from the analysis. Figure 1 shows a flowchart of patient inclusion. An overview of the participating centres and the number of BA patients from each centre is shown in Table 1.

### 3.1. Number of IBA Patients before and during the Lockdown Period

Overall, prior to the lockdown, a median of 6 (range: 1–15) patients who acquired IBA were born per centre per month, versus 7 (6–9) during the lockdown period (*p*= 0.34, see Table 2). 

In countries with national BA data a median number of 4 (1–13) and 3 (2–3) patients who acquired IBA were born in the pre-lockdown and lockdown period, respectively (*p* = 0.14). In centres from countries with part of national BA data, there was an increase in the median number of IBA patients when comparing the pre-lockdown and lockdown period: 2 (0–7) and 4 (4–6), respectively (*p* = 0.01). 

The median number of IBA patients born per month decreased strongly but was not statistically significant in the Netherlands (0.5 (0–3) vs. 0, 100% decrease, *p* = 0.11) and in the United Kingdom (3 (0–11) vs. 1 (1–3), 41.6% decrease, *p* = 0.17). Data from other countries/centres are shown in Table 2 and Figure 2.

When we only considered countries with national BA data, 272 IBA patients from five countries, born between 1 January 2015 and 31 May 2020 were included. During the lockdown, the overall odds of acquiring IBA were 0.68 (95% CI: 0.34–1.37, *p* = 0.28) compared to the pre-lockdown period. The OR of acquiring IBA during the lockdown period compared to the pre-lockdown period decreased in the Netherlands (OR: 0.25 [95% CI: 0.01–4.03, *p* = 0.33]) and United Kingdom (OR: 0.64 [95%CI: 0.26–1.57, *p* = 0.33]); however, this failed to reach statistical significance (Table 3). An overview of the incidence of IBA in different time periods is shown in Figure 3.

### 3.2. Association between Patients Acquiring IBA and Strictness of Lockdown

Appendix A shows the association between the SI and the percentage change of mean number of IBA patients born per month. When all centres were included, no statistically significant association was reached (B = −0.33, *p* = 0.31). Due to confounding factors in Padova and in Hannover (i.e., significant changes in referral patterns leading to more IBA patients being referred to the hospital than usual, and a steep increase in the number of cases), we also performed an analysis without the data of these centres. The association between the percentage change of mean number of IBA patients born per month per centre and SI then reached statistical significance (B = −0.73, *p* = 0.03) (Figure 4 and Appendix A).

For countries with national BA data only, Spearman’s correlation coefficient between percentage change of mean number of IBA patients born per month per centre and SI was −0.60 (*p* = 0.35). Logistic regression analysis yielded an OR of 0.99 (95% CI: 0.98–1.00, *p* = 0.25) per added SI point on IBA incidence during the period of the strict lockdown. For example, compared to the pre-lockdown period, the OR of acquiring IBA in the United Kingdom during the lockdown period was 0.65 (Appendix A).

When IBA incidence during COVID lockdown was compared to the IBA incidence from 1 March–31 May from 2015 up to and including 2019, every 1-point increase in the SI decreased the odds of acquiring IBA; however, this was not statistically significant (OR: 0.996 [95% CI: 0.98–1.01, *p* = 0.54]). 62 (23%) IBA patients were included for this analysis.

### 3.3. Age at KPE

There was no significant difference in median age at KPE (51 vs. 53, *p* = 0.73) between the pre-lockdown period and lockdown period overall, or in countries with national and part of the national BA data (48 vs. 57, *p* = 0.29 and 58 vs. 53, *p* = 0.43, respectively, see Table 4).

## 4. Discussion

The aim of this study was to investigate the incidence of IBA in various European countries before and during the COVID-issued lockdown in 2020. Furthermore, we aimed to investigate if the strictness of the lockdown had an effect on the change in IBA incidence from the pre-lockdown period to the period of strict lockdown. Our results indicated that overall, there was a variable change in the incidence of IBA during the COVID-issued lockdown in comparison to the period prior to the lockdown. 

We observed a negative association between stringency index and the percentage change of mean IBA patients born per month per centre. This finding indicated that a stricter government issued COVID-lockdown in a country was accompanied by a simultaneous larger decrease or even absence of the birth of patients with IBA in the period in which this lockdown was issued. 

These results should be interpreted with caution. Incidence most accurately displays how common a disease is. In our study, incidence could not be calculated in countries where complete national data were not included because a reliable population at risk (i.e., live births in a region or country) could not be determined. To include centres from countries with part of the national BA data, we deemed it justifiable to use an alternative method with the number of IBA patients born per month. This method was based on two assumptions: an equal number of live births per month (not true by definition) and a consistent referral pattern of first- and second-line healthcare. 

A change in referral pattern can lead to an over- or underestimation of the number of BA patients. A change in referral pattern occurred in two centres and they were excluded from the analysis. There was no change in referral pattern in other countries with part of the national BA data. 

Furthermore, to exclude unnoticed bias due to changes in referral pattern, we analysed the effect of the COVID lockdown on IBA incidence separately in countries with national BA data. We again observed a negative association between SI and mean number of IBA patients born per month as well as incidence. However, these observations failed to reach statistical significance. We therefore have to stress that although this study found a negative association between stringency index and the percentage change of mean IBA patients born per month per centre, our results should be interpreted with caution and our conclusion cannot be a final one due to the limitations of our used method. 

Studies investigating the epidemiology of (I)BA in Europe and around the globe have been performed in different countries. Results were conflicting with regard to the existence of clustering in space or time of the birth of BA patients [18,19,20,21,22,23,24,25,26,27,28]. Even in studies where clustering was observed, discrepancy existed in relation to the time of year in which BA clustered and whether BA clustered in rural or urban areas. Rarity of BA leads to epidemiological limitations, especially when most studies investigating clustering are not able to incorporate an explanatory variable for clustering in their analysis. 

When most European countries went into a COVID lockdown, aiming to reduce transmission of COVID-19 infections in March 2020, it provided us with an ‘experiment’ on a European scale: a period of three months in which transmission of other infectious pathogens was also reduced during the lockdown in various countries, allowing us to investigate the effect of this reduction on BA incidence. This is one of the main strengths of our study. Furthermore, our study included data on BA patients from many different European countries, which in itself is a rarity in research on BA. However, in order to perform reliable epidemiological on BA epidemiology in Europe (or other international regions), complete national databases should preferably be used to include all BA patients and have a defined population at risk (i.e., live births). Furthermore, it would be of interest to investigate the relation between non-COVID infection rates in the population and incidence of IBA in Europe to further investigate an infectious aetiology of (I)BA. 

During the lockdown, BA patients were diagnosed and treated later than before the lockdown due to logistic reasons which was illustrated by a higher age at KPE in our study, leading to a long absence of BA patients presenting to the hospital. By analysing the month of birth of patients who acquired IBA instead of the month of presentation, we negated any effect of late presentation on the analysis. However, this method assumed that the (viral) insult proposedly leading to IBA took place perinatally. The timing of the insult leading to the occurrence IBA is still unknown and the insult may occur prenatally [29]. For this study, however, we assumed the insult proposedly leading to IBA took place around the birth of the patient. Our results did not exclude a prenatal insult as cause for IBA. 

Length of the lockdown was defined as a relative short time period of three months, in which few BA patients were born. We deemed this short period appropriate as extending the period would include the relaxation of restrictions, possibly confounding our data. 

Developmental or genetic causes can also still very well be involved in the pathophysiology [9,30,31]. Furthermore, lockdown measures were primarily aimed to reduce transmission of air-borne viral particles, while infection via contaminated food or ingestion of a toxin such as biliatresone (known to cause BA in animals) may still have occurred despite a strict lockdown [32,33]. These factors should be kept in mind when interpreting our results. 

Despite the limitations in this study, collaborative projects as these could aid in tackling the issue of the rarity of a disease in epidemiological, clinical and mechanistical studies, allowing for a greater number of included patients, thereby having more potential to unravel conundrums. It would be of interest to investigate BA incidence in the years after the COVID lockdown to evaluate the potential sustained consequences of an impactful lockdown on the incidence of (I)BA.

## 5. Conclusions

In conclusion, this collaborative study between thirteen European centres investigated the effect of the strict COVID lockdown that was issued in various countries throughout Europe from March 2020. We found an association between the percentage change of mean IBA patients born per month and SI, an index for lockdown strictness, indicating that a stricter COVID lockdown in a country was accompanied by a simultaneous larger decrease or even absence of patients born with IBA in the period in which the lockdown was issued. Our analysis was limited by assumptions and limited statistical power. Furthermore, in countries with complete national BA data the association between IBA incidence and SI failed to reach statistical significance. Our results should therefore be interpreted with caution. Our findings may suggest an environmental factor at play in the pathophysiology of IBA. We hope that this study, using the European Reference Network RARE-Liver, will be the starting point of future collaborative efforts to elucidate the pathophysiology of this rare disease.

## Figures and Tables

**Figure 1 jcm-12-00775-f001:**
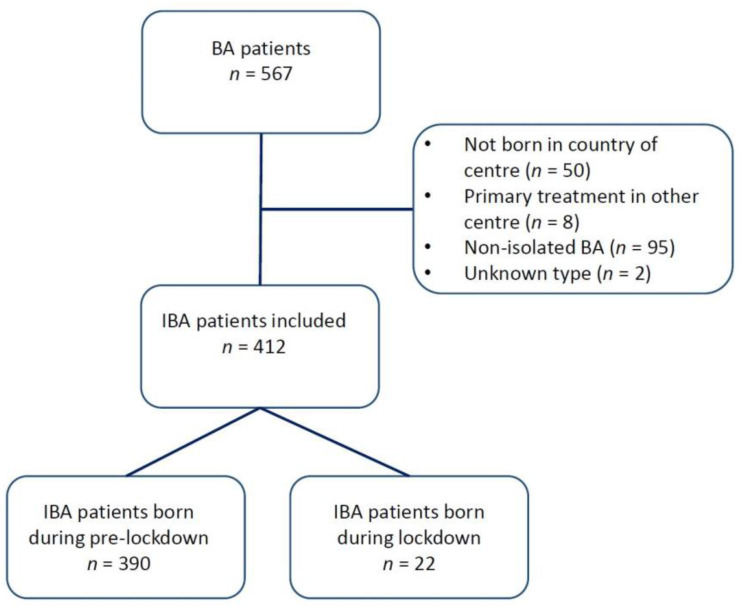
Flowchart of patients included in this study.

**Figure 2 jcm-12-00775-f002:**
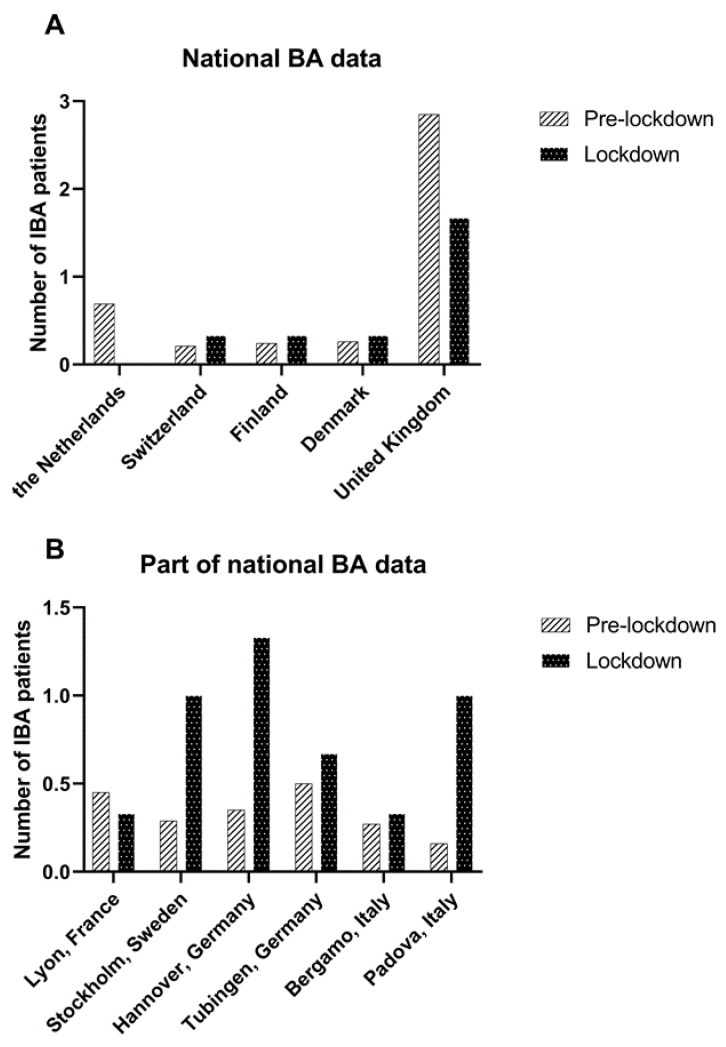
Mean number of IBA patients treated per month in the pre-lockdown period (January 2015 –March 2020) and lockdown period (March 2020–June 2020) in (**A**) countries with national BA data and (**B**) in centres from countries where a part of the national BA data was available.

**Figure 3 jcm-12-00775-f003:**
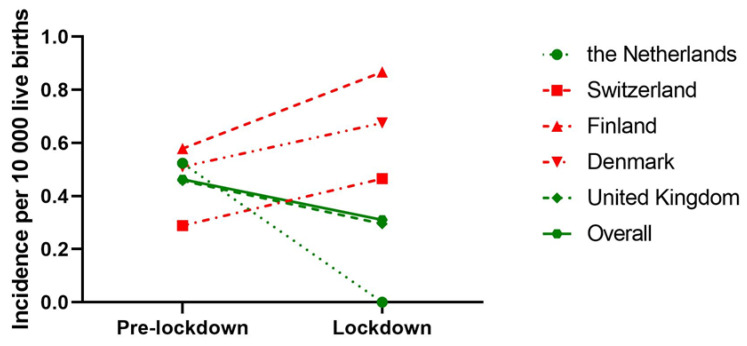
Incidence of IBA prior (January 2015–March 2020) and during the lockdown (March 2020–June 2020) of countries where BA care is centralised. Incidence is expressed as number of IBA patients per 10,000 live births. Different symbols represent different countries and dashed lines represent countries, while the connected line represents overall incidence of all countries. Green lines indicate a decrease in IBA incidence from pre-lockdown to lockdown period, while a red line indicates an increase.

**Figure 4 jcm-12-00775-f004:**
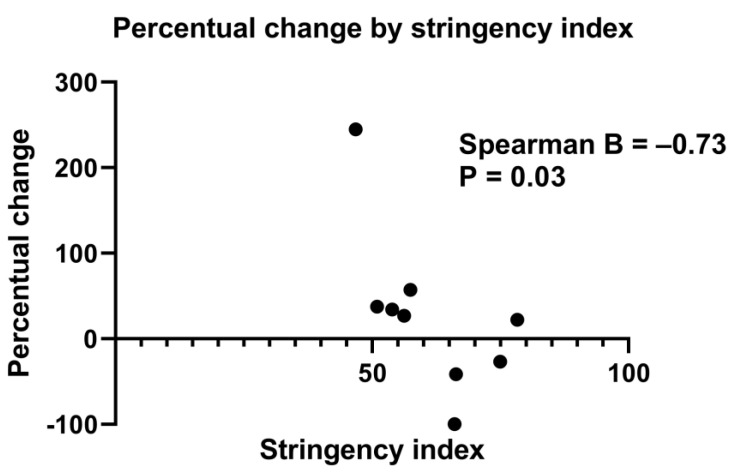
Scatter plot showing the percentage change of mean IBA patients born per month (from pre-lockdown to lockdown period) for every stringency index of the participating centres. *n* = 9, Hannover and Padova were excluded (three centres from United Kingdom were merged into one). Spearman’s correlation coefficient = −0.73, *p* = 0.03.

**Table 1 jcm-12-00775-t001:** An overview of the participating centres and the number of BA patients that were treated in each centre and born between 1 January 2015 and 31 May 2020. A distinction is made between IBA and non-IBA patients. IBA patients were included in this study.

Centre	Number of BA Patients		
	IBA	Non-IBA (% of Total)	Overall
Total	412 (80)	95(20) *	486
Groningen, the Netherlands	43 (88)	6 (12)	49
Geneva, Switzerland	14 (78)	4 (22)	18
Helsinki, Finland	16 (70)	7 (30)	23
London, England	76 (78)	21 (22)	97
Leeds, England	59 (91)	6 (9)	65
Birmingham, England	47 (78)	13 (22)	60
Copenhagen, Denmark	17 (89)	2 (11)	19
Stockholm, Sweden	21	NA	21 *
Lyon, France	29 (81)	7 (19)	36
Padova, Italy	13 (100)	0 (0)	13
Hannover, Germany	26 (53)	23 (47)	49
Tübingen, Germany	33 (87)	5 (13)	38
Bergamo, Italy	18 (95)	1 (5)	19

* Stockholm provided data on IBA patients only. The proportion of non-IBA patients was calculated without data from Stockholm in this Table.

**Table 2 jcm-12-00775-t002:** Median number of IBA patients born per month per participating centre or country (national BA data).

Centre	Median Number of IBA Patients Born Per Month (Range)				
	Pre-Lockdown	Lockdown	% Change	*p*-Value	Stringency
	Countries with national BA data				
Netherlands	0.5 (0–3)	0 (0–0)	−100	0.11	66.1
United Kingdom *	3 (0–11)	1 (1–3)	−41.6	0.17	66.4
Denmark	0 (0–2)	0 (0–1)	26.9	0.66	56.2
Finland	0 (0–2)	0 (0–1)	37.5	0.69	51.0
Switzerland	0 (0–2)	0 (0–1)	57.1	0.57	57.5
Overall national	4 (1–13)	3 (2–3)	−37.3	0.12	
	Centres with part of national BA data				
Lyon, France	0 (0–2)	0 (0–1)	−26.7	0.74	75.0
Bergamo, Italy	0 (0–3)	0 (0–1)	22.2	0.70	78.3
Tübingen, Germany	0 (0–3)	1 (0–1)	34	0.46	53.9
Stockholm, Sweden	0 (0–2)	1 (1–1)	244.8	0.01	46.8
Hannover, Germany	0 (0–3)	0 (0–4)	280	0.61	53.9
Padova, Italy	0 (0–1)	1 (0–2)	525	0.02	78.3
Overall, partly national	2 (0–7)	4 (4–6)	130	0.01	
Overall	6 (2–15)	7 (6–9)	16.6	0.36	

*p*-values were generated using the Mann–Whitney *U* test. Pre-lockdown period was compared with the lockdown period. Green colour represents a percentage decrease in mean number of IBA patients compared to the pre-lockdown period while red indicates an increase. SI is also displayed; colours represent a range of SI where green is the relatively highest and red the relatively lowest. * = In the United Kingdom, data from London, Leeds and Birmingham were combined since treatment is centralised to these three centres.

**Table 3 jcm-12-00775-t003:** Results of logistic binominal regression analysis of IBA incidence prior and during the strict COVID-lockdown per participating country. Odds ratios (OR) are shown which indicates the odds of acquiring IBA during the strict COVID-lockdown, using the odds of acquiring IBA prior to the COVID-lockdown as reference category. Colours represent a range of stringency index where green is the relatively highest and red the relatively lowest.

Country		Odds Ratio (95% Confidence Interval)			Total IBA Patients
	Stringency Index	Pre-Lockdown	Lockdown	*p*	*n*
Overall		ref	0.68 (0.33–1.35)	0.27	272
The Netherlands	66.1	ref	0.25 (0.01–4.03)	0.33	43
United Kingdom	66.4	ref	0.64 (0.26–1.56)	0.33	182
Denmark	56.2	ref	1.32 (0.18–9.98)	0.79	17
Finland	51.0	ref	1.50 (0.20–11.35)	0.70	16
Switzerland	57.5	ref	1.61 (0.21–12.30)	0.65	14

**Table 4 jcm-12-00775-t004:** Median age at Kasai portoenterostomy (KPE) during the pre-lockdown period (January 2015–March 2020) and lockdown period (March 2020–June 2020).

	Median Age at KPE in Days (Range)		*p*-Value
	Pre-Lockdown	Lockdown	
	Countries with national BA data		
Groningen, The Netherlands	55.5 (27–132)	NA	NA
Geneva, Switzerland	58.5 (33–90)	52 ^†^	0.59
Helsinki, Finland	64 (13–154)	84 ^†^	0.45
London, England	49 (16–127)	63 (62–64)	0.09
Leeds, England	41 (9–145)	53 (41–65)	0.37
Birmingham, England	41.5 (22–115)	32 ^†^	0.15
Copenhagen, Denmark	43 (16–148)	45 ^†^	0.83
Overall centralised	48 (9–154)	57 (32–84)	0.29
	Countries with part of national BA data		
Lyon, France	51 (17–179)	31 ^†^	0.34
Padova, Italy	79 (28–92)	51 (35–126)	0.64
Hannover, Germany	54 (29–142)	59 (54–70)	0.43
Tübingen, Germany	57 (24–107)	23 (19–27)	0.03
Bergamo, Italy	67 (41–101)	105 ^†^	0.10
Stockholm, Sweden *			
Overall non-centralised	57.5 (17–179)	52.5 (19–126)	0.43
Overall	51 (9–179)	53 (19–126)	0.73

* Stockholm provided number of patients born per month, age at KPE could not be calculated. ^†^ One patient was born in this time period and underwent an KPE, therefore no age range was provided.

## Data Availability

The data that support the findings of this study are available from the corresponding author upon reasonable request.

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
