# Peer review of "Incidence of Isolated Biliary Atresia during the COVID Lockdown in Europe: Results from a Collaborative Project by RARE-Liver"

_jcm, 2023, doi:10.3390/jcm12030775_

Round 1
Reviewer 1 Report
“This study showed a decrease in Isolated Biliary Atresia patients born from lockdown, a major behavioral restriction for COVID-19.”
This is a very interesting paper to me, but I have a few questions and suggestions.
① Text Line 110 KPE → Kasai portenterostomy (KPE)
② The Discussion fully explains the limitation of this study. Therefore, data analysis with less confusion is better.
Please consider the following
Text Line 141 Is it necessary for two groups?
Is "centers with part of the national data" necessary in the process of drawing the conclusions of this study?
About table 2、
Countries with national BA deta to Centre, and data for United Kingdom are shown for London, Leeds, and Birmingham respectively. I propose to remove the centres with part of the national data.
Please change tables 3,4 and figure 1,2,3 accordingly.
③ Isn't the time period shown in Table ‘Jan 2015-Feb 2020’ and ‘Mar 2020-May 2020’? Or represent it as lockdown and pre-lockdown.
④ Why is there centres with no range in the lockdown data in table 4?
⑤ Did CMV and other infections change before and after lockdown? Is there any data on this?
Author Response
This study showed a decrease in Isolated Biliary Atresia patients born from lockdown, a major behavioral restriction for COVID-19.”
This is a very interesting paper to me, but I have a few questions and suggestions.
① Text Line 110 KPE → Kasai portenterostomy (KPE)
Agreed and adapted
② The Discussion fully explains the limitation of this study. Therefore, data analysis with less confusion is better.
Please consider the following
Text Line 141 Is it necessary for two groups?
Is "centers with part of the national data" necessary in the process of drawing the conclusions of this study?
About table 2、
Countries with national BA deta to Centre, and data for United Kingdom are shown for London, Leeds, and Birmingham respectively. I propose to remove the centres with part of the national data.
Please change tables 3,4 and figure 1,2,3 accordingly.
We agree with the reviewer that some confusion accompanies the division of the data into two groups. However, we feel that the distinction is necessary. In short, we feel that all centres are needed to be included to draw adequate conclusions. These centers also display the uniqueness of this European data set in which as many centers from as many different countries as possible have participated. This basically is the first European study into BA within the RARE-LIVER framework. Therefore we would prefer to include all centers. Furthermore, our conclusion is supported by an analysis performed by using data from all centres (2 centres excluded due to confounding factors). As our methods are limited by the factors mentioned in the discussion, we feel that the distinction is necessary and a more accurate, additional sub-analysis is required (only using national BA data). We have adapted the manuscript to clarify this distinction. To clearly outline which are national data, we added a sentence in the method section (line 171). However, given the arguments above, we would prefer to keep the inclusion of all participating centres.
③ Isn't the time period shown in Table ‘Jan 2015-Feb 2020’ and ‘Mar 2020-May 2020’? Or represent it as lockdown and pre-lockdown.
Agreed and adapted to ‘Pre-lockdown’ and ‘Lockdown’
④ Why is there centres with no range in the lockdown data in table 4?
There are centres where only one patient was born during the lockdown and/oronly one patient underwent a KPE. Therefore, no range was provided in table 4. We have added ‘†’ in the table and explained in the figure legend that no range is provided in the figure legend.
⑤ Did CMV and other infections change before and after lockdown? Is there any data on this?
The reviewer raises an important point as this is an essential question to further substantiate an infectious origin of isolated biliary atresia. Unfortunately, data on any other infections than COVID (CMV included) are unavailable for this particular study. In the Netherlands, there was overall a decreased number of viral and other infections during the COVID-lockdown, especially in children (cited in the manuscript). We have added a section in the discussion addressing this point (line 434).
In our study population there is no data on CMV positive versus negative BA cases before and during lock-down as CMV PCR is 1) not standard of care for BA in all centres and 2) identifying CMV status would have required personal patient data which did not fall under the scope of the study nor on the available ethics approval.
Reviewer 2 Report
The aim of this study was to investigate the incidence of isolated BA (IBA) in various European countries before and during the COVID-issued lockdown and to assess the association between strictness of lockdown and IBA incidence. Results showed there was an inverse association between stringency index and percentage change in IBA.
Strength:
It might be the first study to investigate the correlation between IBA and COVID-19 pandemic
Weakness
Retrospective study
Comments in details;
1. A detailed inclusion and exclusion criteria for enrolled subjects should be added in the Materials and Methods part. Also, a patient select flow chart is needed.
2. Is the inclusion and exclusion criteria for each nation constant?
3. The proportion of BA and non-BA patients in different countries is dramatically different. Why?
4. The authors need to clearly illustrate how the stringency index (SI) was calculated and how many items was included for calculating SI.
5. What about the COVID-19 infection rates in different countries during the same period of time? I think the COVID-19 infection rate is another very important parameter to measure stringency index.
6. How many parents of included infants have been infected by COVID-19 during the pandemic? Have any of included infants been infected by COVID-19? If yes, how many?
Author Response
The aim of this study was to investigate the incidence of isolated BA (IBA) in various European countries before and during the COVID-issued lockdown and to assess the association between strictness of lockdown and IBA incidence. Results showed there was an inverse association between stringency index and percentage change in IBA.
Strength:
It might be the first study to investigate the correlation between IBA and COVID-19 pandemic
Weakness
Retrospective study
Comments in details;
- A detailed inclusion and exclusion criteria for enrolled subjects should be added in the Materials and Methods Also, a patient select flow chart is needed.
A flow chart has been included in the manuscript as figure 1. Inclusion and exclusion criteria were added in the manuscript.
- Is the inclusion and exclusion criteria for each nation constant?
All cases with BA are registered in the centralized countries, in this way the inclusion and exclusion criteria for each country were constant as the changes diagnostic workup and/or definition of BA remained unchanged during the study period. We have not performed any analysis into how the diagnosis of BA was established, we have trusted the consistency of the centers/countries databases (which are, in most cases, internationally well-known databases from which many studies have been produced).
- The proportion of BA and non-BA patients in different countries is dramatically different. Why?
The mean percentage of non-IBA patients is 18, which is quite well within the internationally reported range in the literature. Looking at the data there are 2 outliers (0 47%, respectively) – which we cannot explain except for statistical coincidence (which can easily occur in this rare disease).
- The authors need to clearly illustrate how the stringency index (SI) was calculated and how many items was included for calculating SI.
This was added from line 119 to 128.
- What about the COVID-19 infection rates in different countries during the same period of time? I think the COVID-19 infection rate is another very important parameter to measure stringency index.
COVID-19 infection rate is an important parameter to measure the strictness of a lockdown as well as the effectiveness. However, there are several limitations of using this method. Especially in the first part of the pandemic, not everybody was tested in every country and testing capacity was not sufficient in every country. Therefore, we did not find number of COVID-19 infections a reliable parameter of stringency for this study specifically. Furthermore, the number of daily COVID-19 infections underwent much fluctuation throughout the 3 months of lockdown. By taking an average or median of daily COVID-19 infections over the 3 month period, these fluctuations are negated while they can have important consequences. We have added a sentence in the methods section, addressing this point (line 143)
- How many parents of included infants have been infected by COVID-19 during the pandemic? Have any of included infants been infected by COVID-19? If yes, how many?
We thank the reviewer for this interesting and relevant question. Unfortunately, it cannot be answered with the data at hand. Answering this question is beyond the scope of the paper and of the available ethical approvals.
Round 2
Reviewer 1 Report
The areas pointed out by the review have been carefully corrected.
Report 1:②
I understand that ‘national BA data’ and ‘part of national BA data’ are needed.
I accept the paper as published.